# Feminization of the precarious at the UNAM: Examining obstacles to gender equality

**Lu Ciccia[1]\*, Geraldine Espinosa-Lugo[2], Graciela Garcia-Guzman[3], Jaime Gasca-Pineda[4], Patricia Velez[5], Laura Espinosa-Asuar** [4]¤\*

1 Centro de Investigaciones y Estudios de Género, Universidad Nacional Autónoma de México, México City, México, 2 Facultad de Filosofía y Letras, Universidad Nacional Autónoma de México, México City, México, 3 Secretaría Técnica, Instituto de Ecología, Universidad Nacional Autónoma de México, México City, México, 4 Departamento de Ecología Evolutiva, Instituto de Ecología, Universidad Nacional Autónoma de México, México City, México, 5 Departamento de Botánica, Instituto de Biología, Universidad Nacional Autónoma de México, México City, México

¤ Current address: Departamento de Microbiología Molecular, Instituto de Biotecnología, Universidad Nacional Autónoma de México, Cuernavaca, Morelos, México
* lu_ciccia@cieg.unam.mx (LC); lauasuar@ecologia.unam.mx (LEA)

## Abstract

The STEM workforce is marked by the persistent underrepresentation of women. Herein, we seek a better understanding of this gender gap in different science disciplines within Latin America. Specifically, we analyzed a case study: the professional development of women in science research institutes of the National Autonomous University of Mexico (UNAM). This interdisciplinary work analyzed quantitative and qualitative data through an intersectional philosophical lens, employing specific analytical tools drawn from feminist epistemology. We examined the interplay between horizontal and vertical segregation, symbolic and structural obstacles, and economic labor precariousness within the framework of gender norms. Shared trends in the Global North were analyzed to understand the perpetuation of gender stereotypes in the production of scientific knowledge. Additionally, we examined the relationship between the values embedded in gender norms and the cultural capital--here defined as encompassing both economic status and social legitimacy--associated with each discipline. Our findings indicate that, although women are underrepresented in pSTEM, they experience less vertical segregation than their counterparts in STEM related to the Natural Sciences. This suggests that knowledge areas currently associated with the highest cultural capital (pSTEM) may impose primarily symbolic rather than structural barriers for women. By contrast, in fields characterized by less masculine-coded values, women appear to face predominantly structural obstacles, as evidenced by the vertical segregation observed. These results contribute to a deeper understanding of the gender biases that exclude women from STEM disciplines.

**Data availability statement:** All relevant data are within the manuscript and its Supporting Information files.

**Funding:** This work was supported by Dirección General de Asuntos del Personal Académico, Universidad Nacional Autónoma de México (award number: PAPIIT IA400223; https://dgapa.unam.mx/index.php/impulso-a-la-investigacion/papiit). This grant also funded the scholarship of GEL. The funders had no role in study design, data collection and analysis, decision to publish, or preparation of the manuscript.

**Competing interests:** The authors have declared that no competing interests exist.

## Introduction

### STEM and pSTEM disciplines: analytical context

The acronym STEM (Science, Technology, Engineering, and Mathematics) was introduced to the United States by the National Science Foundation in the year 2000 to encourage interest in the disciplines it encompasses [1]. This promotion arises from the increasing demand for scientific and technological education, driven by the constant changes in technology and the growing digitalization of contemporary societies [2]. Nonetheless, workforce representation in STEM is characterized by a strong gender disparity, with the under-representation of women [3].

Systematic research addressing the underrepresentation of women in science and technology predates the formalization of the STEM acronym [3]. In recent years, there has been a marked increase in studies examining the challenges and barriers faced by women in science. For instance, over the last decade, around 15,000 articles covering various disciplines, including psychology, economics, biology, physics, education, mathematics, and others, have been published [4]. In the context of this disciplinary diversity, the gender gap within STEM has become especially notable due to the 'horizontal segregation' (fewer women in occupational areas considered 'masculine') that characterizes these disciplines. Given the high demand for STEM workers [5] and the focus on "digitally driven economies" [2], socioeconomic status intersects with the STEM gender gap.

Herein, we distinguished between STEM and pSTEM disciplines to specify a subset of STEM fields, primarily focusing on physics [6], and to address the gender gap in physics-related disciplines, differentiating it from other STEM sciences where women might have more equitable representation. This approach acknowledges that the challenges and barriers women face may differ across various STEM fields.

Published literature on women in pSTEM is still insufficient to attain a comprehensive analysis of gender bias in scientific disciplines. Specifically, the relationship between women and STEM in Latin America has been neglected, with some exceptions [7,8]; and comparative analyses aiming to understand differences between STEM departments in Latin American universities remain to be elucidated. This approach is essential for two main reasons: 1) to contribute to a better understanding of the gender gap regarding the reality of women in the different disciplines within STEM in Latin America, and 2) to analyze commonalities regarding the situation in the Global North. These points are essential in understanding how gender norms operate to exclude women in specialized academic contexts.

Our research efforts focused on the National Autonomous University of Mexico (UNAM). Previous works at this university [9] have demonstrated gender disparity regardless of the STEM disciplinary area. In addition, the academic careers of women in this university face significant obstacles in contrast to men who achieve seniority in a shorter period, meaning that female researchers are less

represented at the highest levels of appointment and in the field of scientific research in general. The combination of both facts results in a marked segregation of women [9].

**An exploration of the obstacles underpinning gender equality**

This case study aims to untangle the gender gap in research institutes of the UNAM that belong to the fields of Exact and Natural Sciences. We conducted a comparative analysis to understand how horizontal segregation and vertical segregation (the absence of women in senior posts) interact, and, at the same time, their relationship with the income level of academic personnel. To do this, we selected six representative pSTEM institutes, and eight institutes of STEM, where, due to the increasing number of women, less horizontal segregation is observed (we will refer to those other areas that are not pSTEM simply as STEM, understanding that this is a broader category that we will use to encompass some Natural Sciences unrelated to physics and similar areas). Additionally, we took on the task of collecting more specific data from one of the six pSTEM institutes (Institute of Mathematics (IM), and two of the eight institutes (Institute of Biology and the Institute of Ecology (IB and IE)) to provide a more comprehensive overview.

Traditional views assume that women face numerous disadvantages in institutions where they are underrepresented. In contrast, we hypothesize that although there are few women in the pSTEM, they are less affected by vertical segregation than STEM women. To test this hypothesis, we analyzed vertical segregation regarding positions within academic and incentive programs at selected STEM and pSTEM institutes.

This hypothesis is grounded in the observation that pSTEM fields are often seen as embodying values traditionally associated with masculinity, such as objectivity, neutrality, abstraction, reason, and universality, having the highest economic and social legitimacy [10–12]. We will refer to this combination of economic and social legitimacy as 'cultural capital' [13]. Particularly, we described that the disciplines that most embody values associated with masculinity are those that are more highly valued [13,14] or possess greater cultural capital. In line with this description, it has been shown that values typically associated with femininity, such as emotion, sensitivity, and empathy, are often excluded, or even regarded as obstacles to the production of scientific knowledge [14–16]. These gendered values are located within a symbolic dimension [10,17]. In this dimension, the values embodied in the practice of science matter: if those values are associated with masculinity--as is paradigmatically the case in pSTEM--those who do not align with them, typically women, will encounter symbolic barriers. That is, not explicit prohibitions based on economic or social conditions, but symbolic ones, which assume that 'they' do not identify with or feel attracted to those disciplines. We propose that once these barriers are partially subdued, women's progress is comparable to that of men, because they become experts in disciplines that possess high social legitimacy. In other words, horizontal segregation could mitigate vertical segregation in pSTEM disciplines, characterized by embodying a symbolic barrier for women [12].

This case study examines in detail the relationship between segregation and the values associated with pSTEM and STEM disciplines, through an intersectional lens grounded in feminist epistemology. The results indicate that gender distribution in academic positions and incentive programs is comparable in pSTEM institutions, whereas in STEM institutes, significant disparities persist, with men predominating in higher ranks and women more prevalent in lower positions. In other words, reducing horizontal segregation in fields like ecology and biology could exacerbate vertical segregation. We propose that this occurs because, while these fields may be less masculinized (valuing direct interest in life and the environment), the resulting knowledge often carries less economic and social legitimacy (cultural capital). Consequently, gender inequality persists as men tend to occupy decision-making positions, --entailing social status and economic income-- [4,13,18], reinforcing the idea that epistemic authority in these fields is tied to the academic status of researchers. Specifically, reduced horizontal segregation can reveal clearer structural obstacles (concrete barriers stemming from gender norms and stereotypes) that hinder the advancement of female staff to senior appointments, even when equally qualified. These structural barriers are previously reported for academia, particularly at UNAM [19].

## Methods

### Institutional context: academic structure at UNAM

Academic staff at the research institutes of the National Autonomous University of Mexico (UNAM) are classified into two main categories: research and technical personnel. This case study focuses primarily on research personnel, although some characteristics of the technical academic staff are also briefly addressed.

Two incentive programs were analyzed: the Programa de Primas al Desempeño del Personal Académico de Tiempo Completo or Academic Personnel Performance Incentive Program (PRIDE), an internal performance-based evaluation system at UNAM, and the Sistema Nacional de Investigadoras e Investigadores or National System of Researchers (SNII), a federally managed distinction program. Both are merit-based mechanisms that reward academic productivity through hierarchical ranks and financial incentives. In the case of PRIDE, approximately half of the evaluation committee members are external academics, while for SNII the committees are fully composed of external researchers not affiliated with the institutes under evaluation, contrasting with the internal evaluation bodies responsible for determining academic ranks within each institute.

### Data collection

We gathered data on academic ranks and PRIDE levels (Table 1, S2 File) for research personnel across 14 UNAM research institutes: six categorized as physical STEM (pSTEM) --Mathematics, Astronomy, Engineering, Applied Mathematics and Systems, Nuclear Sciences, and Physics -- and eight as biological STEM --Biology, Ecology, Biomedical Research, Genomic Sciences, Biotechnology, Chemistry, Cellular Physiology, and Geology. All are UNAM-affiliated institutions that exhibit varying gender distributions among their academic staff: pSTEM institutes have higher proportions of masculine personnel compared to STEM institutes (Fig A in S1 File).

**Table 1. Data source list.** Acronyms: Institute of Mathematics (IM), Institute of Astronomy (IA), Institute of Engineering (IEn), Institute for Research in Applied Mathematics and Systems (IR), Institute of Nuclear Sciences (IN), Institute of Physics(IP) and eight STEM (Institute of Biology (IB), Institute of Ecology (IE), Institute of Biomedical Research (IBR), Center for Genomic Sciences (CG), Institute of Chemistry (IC), Institute of Biotechnology (IBt), Institute of Cellular Physiology (ICF), Institute of Geology (IG).

| Data type (total counts, except for remuneration data) | Data source | Related institutes | Requested or accesed date |
|---|---|---|---|
| Academic personnel counts by gender corresponding to 2023 | Requested via the General Transparency Program of the Mexican Government [20] | IA, IEn, IR, IN, IP, IBR, CG, IC, IBt, ICF, IG | 01/08/24 |
| Academic personnel counts by name corresponding to 2023 | Requested via UNAM Transparency [21] | IM, IB, IE | 01/06/23 |
| Academic position by levels, and PRIDE incentive program levels for research personnel corresponding to 2023 | Requested via the General Transparency Program of the Mexican Government [20] | IA, IEn, IR, IN, IP, IBR, CG, IC, IBt, ICF, IG | 01/08/24 |
| Academic position, emeritus data, and PRIDE incentive program levels for research personnel (including emeritus position) corresponding to 2023 | Requested via UNAM Transparency [21] | IM, IB, IE | 01/06/23 |
| SNII incentive program levels for research personnel corresponding to 2023 | Requested via UNAM Transparency [21] | IM, IB, IE | 01/06/23 |
| Historical data (academic position, PRIDE, and SNII incentive programs for research personnel | Requested via UNAM Transparency [21] Direct mail petitions to the administration of the analyzed institutes and to General Direction of Academic Personnel (DGAPA) office | IM, IB, IE | 01/06/23 |
| Remuneration data (academic position, PRIDE and SNII incentive programs) for 2023 | Data available on public pages [22–24] | NA | 01/06/23 |

### Qualitative data

Additionally, a more detailed data collection of research personnel's age, SNII levels, emeritus level, and historical information was conducted for three of the fourteen institutes: the Institute of Biology, the Institute of Ecology, and the Institute of Mathematics (Table 1).

### Gender classification

As gender data are classified as personal information under Mexican law, they are subject to legal protection; however, for the Biology, Ecology, and Mathematics institutes, gender was inferred from individuals' given names, which were publicly accessible because UNAM academic staff are public officials. Classification followed Spanish linguistic and cultural conventions: names ending in -o (e.g., Pedro, Mario) were categorized as masculine, and those ending in -a (e.g., María, Laura) as feminine. Names that did not conform to these patterns were categorized according to conventional usage (e.g., Juan as masculine, Luz as feminine). Gender-neutral names are uncommon in Spanish; thus, this method offers a high level of accuracy. A small number of non-Spanish names were classified based on naming conventions in their cultural context, acknowledging a possible margin of error. This classification reflects naming patterns only and does not assume the individuals' self-identified gender. For the remaining institutes, the data were anonymized, allowing the information to be provided aggregated by gender.

### Data analysis

Graphs depicting the distributions of gender, academic ranks and PRIDE incentive program levels were generated by institute and by STEM/pSTEM category, based on 2023 data; SNII data were included only for three selected institutes.

To enable meaningful comparisons across institutes with varying gender distributions, we calculated the proportional representation (named here as proportional distribution) of each gender across academic ranks and incentive program levels (research personnel data only). For example, if an institute had 20 women researchers, with 5 holding the Titular A rank, 10 Titular B, and 5 Titular C, the corresponding proportions would be 0.25, 0.5, and 0.25, respectively. Raw counts and proportional distributions are provided in S2 File.

To compare the proportional distribution of academic ranks and PRIDE program between STEM and pSTEM institutes, we used the Wilcoxon rank-sum test (excluding emeritus ranks due to incomplete data). To test for differences in gender distribution across ranks and programs within each institute, we applied the Kruskal–Wallis one-way analysis of variance by ranks, followed by Dunn's post hoc paired comparisons in R.

For the historical data from the Institute of Biology (IB), the Institute of Ecology (IE), and the Institute of Mathematics (IM), we estimated the average time to promotion. Promotions were analyzed separately for academic rank and for incentive program levels. For academic rank, we considered the initial hiring year and entry-level rank; for incentive programs, we used the year and level of the first registered award. The time between each promotion was then calculated. We applied the Wilcoxon rank-sum test to evaluate differences between genders for each institute.

Lastly, remuneration data were analyzed. Salaries associated with academic ranks and incentive program levels (PRIDE and SNII) were calculated as percentages relative to the highest-paid academic and incentive program level. All analyses and visualizations were conducted in R (version 4.3) [25] and Microsoft Excel.

This interdisciplinary study analyzes quantitative and qualitative (descriptive) data through an intersectional philosophical lens, employing specific analytical tools drawn from feminist epistemology.

## Results

### Gender distribution

Analysis of the gender composition of academic personnel (research and technical) in the studied institutes (eight STEM and six pSTEM; see Methods) revealed two distinct patterns: STEM institutes exhibited either marginal feminization or

marginal masculinization, whereas pSTEM institutes were consistently and markedly masculinized (Fig A in S1 File). This imbalance establishes the baseline for examining differences in academic rank and incentive programs.

## Academic rank and PRIDE incentive program

The Wilcoxon rank-sum test revealed significant differences ($p < 0.05$) in the academic ranks of Titular A and Titular C among research personnel in STEM institutes (Fig 1, S3 File). In contrast, no significant differences were found for pSTEM institutes or PRIDE data (Fig 2, S3 File). Kruskal–Wallis one-way analysis of variance by ranks was not significant for the distribution of gender across ranks and programs within each institute (data not shown).

## Descriptive data for three institutes

Across the three focal institutes (IB, IE, IM), the mean age of research personnel was comparable (IB = 56.57; IE = 56.81; IM = 56.65). However, gender-specific age gaps were more pronounced in IE and IM (6–7 years) than in IB (≈2 years) (Fig B in S1 File). Notwithstanding these differences, overall age distributions by gender were broadly similar (Fig C in S1 File).

We observed disparities in the proportional distribution at the highest levels of the Academic and Incentive Program (Fig D in S1 File) for IB and IE institutes. The IE exhibited the largest gender gap in the Titular C rank and in the top incentive categories (PRIDE D and SNII III). In contrast, the IM showed near gender parity, although proportions were lower at

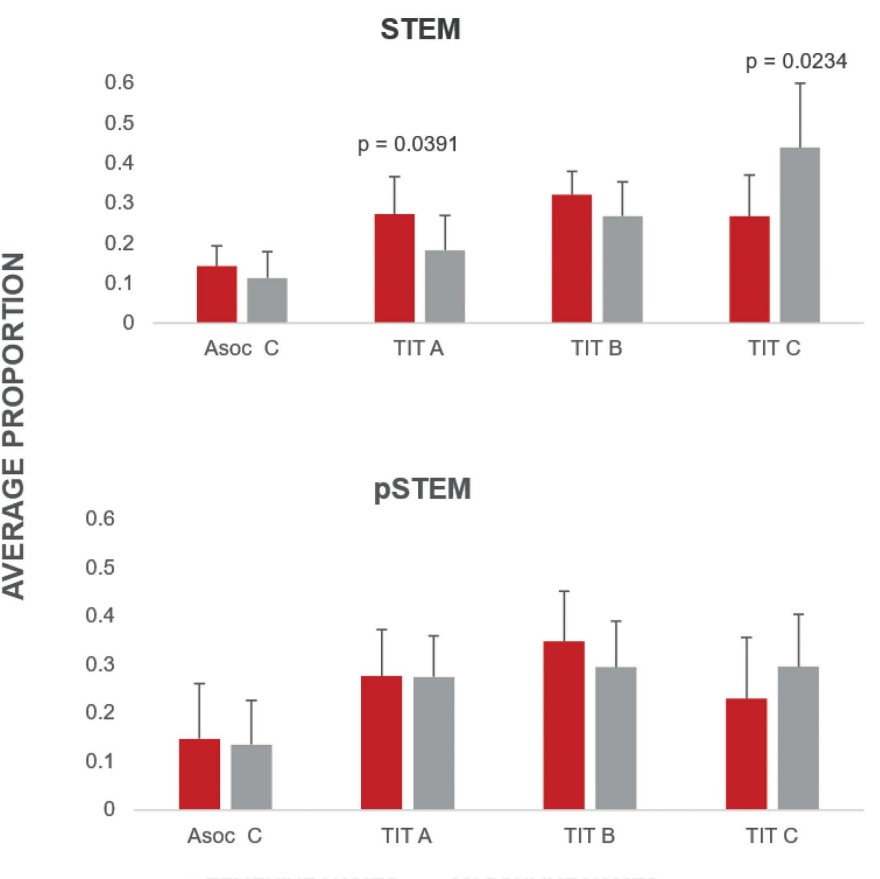

**Fig 1. Average proportional distribution by gender among research personnel in STEM institutes.** Statistically significant differences identified by the Wilcoxon rank-sum test are indicated.

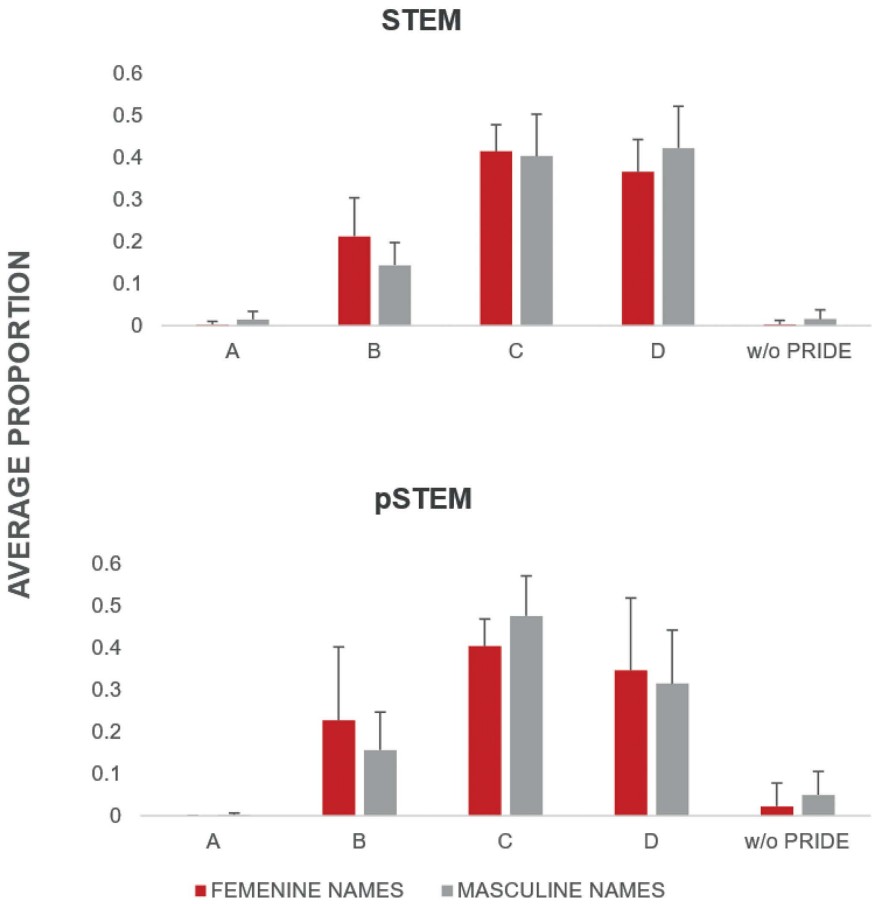

**Fig 2. Average proportional distribution by gender among research personnel in pSTEM institutes.** No statistically significant differences were detected by the Wilcoxon rank-sum test.

SNII III. The honorary Emeritus title was absent among women in both academic rank and SNII, with the sole exception of the SNII in IM (Fig E in S1 File).

### Promotion time and remuneration

Across the three institutes, women generally required more time than men to attain promotions (Fig F in S1 File, S3 File), although some exceptions were noted in the PRIDE and SNII incentive programs. These disparities in progression rates are particularly relevant given the substantial differences in remuneration associated with the highest levels of both academic rank and incentive programs (Fig G in S1 File).

## Discussion

### STEM *vs* pSTEM

The analyzed institutions exhibit diverse gender representation, with varying proportions of women and men: there is a pronounced gender imbalance within the academic personnel of pSTEM institutes, and STEM disciplines such as biology and ecology are not characterized by horizontal segregation (Fig A in S1 File). At the same time, we observed a significant representation of women in the category of academic technical staff in the IE and IB (Table A in S1 File). Remuneration

data (Table B in S1 File) illustrate the economic precariousness and lower social status (referred to here as lower cultural capital) associated with this category. In this line, we observed masculinization of the academic technical staff within the IM (Table A in S1 File). This reflects how the symbolic barriers faced by women in these fields [12] arise even among academic technical staff--a category characterized by lower income and status [26]--yet within an area that holds greater social recognition, highlighting the link between the masculine values that characterize this field of knowledge and the social legitimacy attached to them, whereby what is associated with masculinity is more highly valued [14].

Regarding academic positions and the phenomenon of horizontal segregation, which is not observed in the STEM Institutes, we confirmed the accentuation of their vertical segregation: significant differences in lower and higher ranks for STEM Institutes (Fig 1). In other words, the proportion of women in the highest rank within these institutes (Titular C) is significative lower than men, and the opposite occurs in the Titular A category, one of the lowest ranks: the proportion of women is significantly higher than that of men. Descriptive data for Biology and Ecology confirmed this trend, as well as for Emeritus rank and incentive programs (Pride D, SNII III, and SNII Emeritus) (Fig D and E in S1 File). Given the limited number of observations analyzed separately for each institute, we cannot conclude that the observed trend is significant, as evidenced by the statistical results ($p > 0.05$, data not shown). Nonetheless, using it as descriptive data is valuable because, on the one hand, it reflects the overall patterns observed across all institutes, and on the other, it includes information that is not widely available.

This lower proportion of feminine names in the Titular C position (Fig 1) indicates that structural obstacles could prevail within these disciplines, whose values are less masculinized and, therefore, have a lower level of social prestige compared to pSTEM [12,13]. These structural obstacles tend to be associated with women facing barriers and biases linked to gender-based discrimination [19]. This discrimination is embedded within the symbolic dimension previously discussed. Consequently, women must exert greater effort than their male counterparts to attain comparable promotions, and their professional status tends to receive less recognition [9].

In the case of the pSTEM institutes, where horizontal segregation is evident (Fig A in S1 File), we observed a dilution of vertical segregation. In other words, the proportion of women in the Titular C and PRIDE D is equivalent and without significant differences (Fig 1 and 2). The IM data confirmed this pattern (Fig D in S1 File), as well as for SNII III and SNII Emeritus higher incentives. In SNII Emeritus, there are more women (Fig E in S1 File). These data are consistent with our proposition: when women overcome symbolic barriers and succeed in identifying with disciplines that most embody values associated with masculinity, they gain access to the same positions as men. In other words, they achieve equivalent economic earnings. We observed that structural barriers are diminished when working in occupations of high social prestige.

However, we mention that vertical segregation 'does not disappear', as reflected in higher proportion (although non-significant) Titular C rank, or the absence of women Emeritus academic rank positions, but is greatly diminished compared to STEM institutes' rank. The phenomenon whereby horizontal segregation is related to symbolic barriers has been extensively documented in various gender studies [9,27].

Our observation that vertical segregation is mitigated in pSTEM institutes compared to STEM institutes is consistent with a recent meta-analysis that reviewed numerous articles and found that in the United States and Europe, '...in tenure-track hiring, our national cohort analyses show no increased likelihood that men proceed to tenure-track jobs relative to women in the very fields in which women are most underrepresented (GEMP), although there is a difference in LPS fields' (GEMP = geoscience, engineering, economics, mathematics, computer science, and physical science; LPS = life sciences, psychology/behavioral sciences, and social sciences) [4]. These authors have emphasized the importance of accurately describing gender bias in academic science, as well as acknowledging instances where such bias is absent. In our case, the proportional distribution proved to be a useful measure for identifying the dilution in vertical segregation, which would not have been apparent through a conventional presentation of this type of data, in which pSTEM institutions are typically portrayed as exhibiting imbalanced distributions across academic ranks due to their inherently unequal representation of men and women.

The PRIDE data revealed a pattern aligned with the diminished vertical segregation: although not statistically significant, pSTEM institutes displayed a marginally higher proportion of female names in the PRIDE D category. Given that the evaluation committees for this incentive program include external academics from other UNAM institutions, the absence of significant differences for this data may be linked to anonymous evaluation, reported as an effective strategy for reducing bias in academic assessment [28].

For observed vertical segregation for STEM institutes (Fig 1 and 2, Fig D and E in S1 File), we propose that it could be directly proportional to structural barriers --reflected in the vertical segregation observed-- in these less valued areas (because they are less associated with masculine values and therefore have less cultural capital). The gender hierarchy is reflected in how women are placed in these spaces: women are more precarized because their lower-status posts and incentives result in lower income (Fig G in S1 File). Such female-dominated disciplines have been reported to be associated with lower success rates [29]. In accordance with other studies, we could infer that the pay gap is gendered in the STEM institutes analyzed, since more women are occupying lower-status posts [4,13,18]. In contrast, when obstacles are symbolic (that is, based on the values embodied by disciplines, depending on how strongly they are associated with masculinity), the hierarchy is reflected in the absence of women and male overrepresentation. In these areas of knowledge with significant cultural capital, due to the high social prestige and economic income associated with it [13] the highest posts are less biased for gender reasons: precarization in these cases is shown in the disidentification that women develop concerning these types of disciplines, which are strongly tied to values associated with masculinity. However, we emphasize that vertical segregation also exists in both STEM and pSTEM institutes, but, we observed a lower vertical segregation tendency in pSTEM institutes (Fig 1 and 2; Fig D and E in S1 File). We consider that the context of each institute (e.g., internal evaluation criteria for academic staff), along with many other structural conditions within each institute are also influencing this tendency.

It is important to mention that we have found, in line with previous studies [9], that women generally delay their academic promotion in Biology, Ecology, and Mathematics institutes (Fig F in S1 File). Interestingly, PRIDE IM data was the only significant difference (S3 File). Since promotion time is a personal decision, this data (and the non-significant ones) could reflect a combination of structural and symbolic obstacles. The former obstacle is that women must reconcile their academic-professional life with their family life, a tension not experienced by their male counterparts [30]. This can result in women taking longer to seek rank or level changes, to produce more publications and gain teaching experience, etc. Regarding the symbolic obstacles, women are more insecure and find it harder to trust themselves, a fact that is reinforced by their lower recognition as compared to men [16]. Even in the best case, they must work harder to gain the same recognition as their male counterparts. At this point, which involves subjectivity and qualitative analysis, it is important to investigate how women experience their careers in the three analyzed institutes.

## De-gendering academic fields for equal opportunities

It is crucial to acknowledge that the presence of more women in specific disciplines does not inherently ensure greater equity, as our data from STEM institutes demonstrates. Therefore, promoting equity requires more than simply increasing women's representation in pSTEM fields. Instead, by taking the simultaneity of new materialisms as the basis for our analysis [31], we emphasize the overrepresentation of men and their absence in precarized spaces, both physical and symbolic. For example, policies aimed at including women in science have not been matched by policies for including men in communal roles. This discrepancy explains why, between 1995 and 2013, there was an increase in women entering male-dominated occupations, while there was virtually no change in men's participation in female-dominated fields [13]. This fact reflects that occupations involving communal roles, and associated with femininity, have not been revalorized, and their cultural capital continues to be overshadowed by that of pSTEM.

In our view, effective inclusion policies should address both structural and symbolic obstacles. Specifically, these policies should aim to revalorize communal tasks often associated with femininity and encourage men's participation in these

roles. Achieving this requires challenging the gender stereotypes that feminize certain tasks and masculinize others by associating them with values such as empathy and rationality.

In this case study, we advocate for the contextualization and specificity of each discipline. At UNAM, precarization refers not only to appointments and incentives but also to the distinction between academic technical staff and researchers [26]. One specific policy to address the precarious status of academic technical staff is to revalorize this role in economic, physical, and symbolic terms. We also demonstrated that each institute has unique characteristics influenced by various factors, including the field of knowledge. In this context, stereotypes and forms of precarization also become specific. Although we can make comparisons between institutes, each has distinctive features that deserve qualitative evaluation, and these particularities should be considered in inclusion policies.

Finally, it is important to acknowledge that our interpretation of the observed pattern--namely, greater vertical segregation in STEM institutes compared to pSTEM institutes--may coexist with alternative explanations; nevertheless, we regard it as a reasonably accurate approximation, in line with the studies cited here and the critiques of gender biases from feminist epistemology.

## Conclusions

The STEM institutes related to the Natural Sciences exhibited reduced horizontal segregation but higher vertical segregation compared to pSTEM institutes. We propose that vertical segregation arises from structural obstacles, such as gender-based discrimination, which limit women's opportunities for professional advancement; however, this postulate requires further investigation.

In contrast, we suggest that horizontal segregation in pSTEM is primarily associated with symbolic obstacles faced by women, consistent with previous research in feminist epistemology [9,12,27]. These obstacles are linked to the paradigmatic values upheld in these disciplines--objectivity, neutrality, and universality--which have historically been coded as masculine [10,14].

Our findings indicated that the horizontal segregation in pSTEM institutes is inversely proportional to the phenomenon of vertical segregation, suggesting that when barriers are partially mitigated, the advancement of women becomes broadly comparable to that of men, progressing with fewer impediments as they develop expertise in disciplines endowed with high social legitimacy.

In summary, we emphasize that it is essential to analyze the specificity of the disciplines and their relationship with structural and symbolic obstacles, as well as their interaction with horizontal and vertical segregation phenomena.

We must continue investigating how women may perpetuate stereotypes that distance them from fields like pSTEM while normalizing their presence in feminized precarious spaces, thereby reinforcing their precarization within the academic sphere. The perpetuation of stereotypes often occurs through socialization processes in upbringing and early education, which normalize the gendered attribution of certain values--for instance, the belief that men are inherently more rational while women are more emotional due to their reproductive roles.

Finally, we consider that this is an innovative case study that demonstrates the complex situation of women from science disciplines in a Latin American university. We emphasize that the data we obtained and the interpretation we developed are consistent with findings from certain studies conducted in the Global North [4,13], a fact that reveals the points of convergence between gender norms across different regions of the world and calls for greater dialogue among universities to untangle how androcentric values are reproduced in our modes of knowledge production.

## Supporting information

**S1 File. Supplemental figures and tables.**
(DOCX)

**S2 File. Excel file with raw data and their proportional distributions.**
(XLSX)

**S3 File. Tables reporting p-values from Wilcoxon rank-sum tests.**
(DOCX)

## Acknowledgments

We thank Nagapriya Wright for his impeccable translation and Dr. Teresa Valverde for her valuable statistical advice. We appreciate the work of Camila Ramírez Araujo and Adriana Ruiz Gadea for collecting data.

## Author contributions

**Conceptualization:** Lu Ciccia, Jaime Gasca-Pineda, Patricia Velez, Laura Espinosa-Asuar.

**Data curation:** Geraldine Espinosa-Lugo.

**Formal analysis:** Geraldine Espinosa-Lugo, Graciela Garcia-Guzman, Jaime Gasca-Pineda, Laura Espinosa-Asuar.

**Funding acquisition:** Lu Ciccia.

**Investigation:** Lu Ciccia, Laura Espinosa-Asuar.

**Methodology:** Lu Ciccia, Jaime Gasca-Pineda, Laura Espinosa-Asuar.

**Project administration:** Laura Espinosa-Asuar.

**Supervision:** Lu Ciccia, Laura Espinosa-Asuar.

**Validation:** Lu Ciccia, Laura Espinosa-Asuar.

**Writing – original draft:** Lu Ciccia, Laura Espinosa-Asuar.

**Writing – review & editing:** Patricia Velez.

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
