## [Decision Letter · Decision Letter 0]

11 Jun 2025

Dear Dr. Espinosa-Asuar,

Thank you for submitting your manuscript to PLOS ONE. After careful consideration, we feel that it has merit but does not fully meet PLOS ONE’s publication criteria as it currently stands. Therefore, we invite you to submit a revised version of the manuscript that addresses the points raised during the review process.

We look forward to receiving your revised manuscript.

Kind regards,

Alejandro Botero Carvajal, MD

Academic Editor

PLOS ONE

Journal Requirements:

 [DGAPA, National Autonomous University of Mexico (UNAM) under grant PAPIIT IA400223]. 

[This work was supported by DGAPA, National Autonomous University of Mexico (UNAM) under grant PAPIIT IA400223. This grant PAPIIT IA400223 also funding scholarship of Geraldine Espinosa Lugo and Linda Gisele Zavala Hernández. We thank Nagapriya Wright for his impeccable translation. We appreciate the work of Camila Ramírez Araujo and Adriana Ruiz Gadea for collecting data.]

[DGAPA, National Autonomous University of Mexico (UNAM) under grant PAPIIT IA400223]. 

5. Please expand the acronym “DGAPA” (as indicated in your financial disclosure) so that it states the name of your funders in full.

Additional Editor Comments:

In your Methods section please clearly state how names were determined to be "masculine" or "feminine" and how edge instances were handled.

Reviewers' comments:

Reviewer's Responses to Questions

**Comments to the Author**

1. Is the manuscript technically sound, and do the data support the conclusions?

Reviewer #1: No

Reviewer #2: Partly

2. Has the statistical analysis been performed appropriately and rigorously?

Reviewer #1: I Don't Know

Reviewer #2: No

3. Have the authors made all data underlying the findings in their manuscript fully available?

Reviewer #1: Yes

Reviewer #2: Yes

4. Is the manuscript presented in an intelligible fashion and written in standard English?

Reviewer #1: Yes

Reviewer #2: No

Reviewer #1: This research explores an important topic, and the comparisons across STEM fields are valuable. The differences across discipline identified are interesting, and the explanation provided for those differences plausible. I would recommend some revisions to address some lingering questions and concerns.

First, and most importantly, I would recommend moving the methodology section from the end of the paper, so that it appears prior to the presentation of the results. It's actually hard to understand the findings without having the explanation of what data are being drawn on, and how they are being analysed. The findings will be more comprehensible if you describe what you did ahead of time.

Second, the methodology section can be expanded to provide explanations for key variables, so that people unfamiliar with academia in Mexico can better follow. For example, you provide a brief explanation of PRIDE and SN1, but it was not sufficient for me to understand what these programs are and therefore how to interpret gender differences with respect to them. Some aspects of the methodology are well-explained, but others are less explained.

Third, you provide an explanation for how gender differences vary across discipline, but it wasn't clear that your data allowed you to test or support your explanation. Neither was it entirely clear that there was a robust body of literature that would support your conclusions, over other potential explanations for the differences. Did you have any measure for cultural capital, for example? In the conclusion, you write, "Fields with high cultural capital create significant symbolic obstacles for women, whereas those with lower cultural capital impose structural challenges." It wasn't clear to me what this (and similar) statements are based on. Is there evidence of 'symbolic obstacles' or variations in cultural capital across fields? Why symbolic obstacles (versus discrimination or social closure for example)? It seems like your data is primarily objective data on employment and recognition, with some ability to assess trends over time. However, there didn't seem to be any clear data that allowed you to understand why these differences occur. As a result, I am not convinced that your data entirely supports your conclusions.

Fourth, you might want to also move the 'What is the underlying problem?' section. It is good to establish the 'problem' earlier in the paper. Here too, though, you should be careful not to declare that your findings support conclusions about structural and symbolic barriers, when it is not clear on what variables or analyses these conclusions are based. If you do have evidence of structural and symbolic barriers in academia, please be clearer about what these are.

Thanks for an interesting study.

Reviewer #2: 1-All comparisons are presented descriptively (proportions and means) without significance testing. Incorporate appropriate tests such as chi-square for rank/incentive distributions, t-tests or nonparametric equivalents for promotion intervals, and report p-values or confidence intervals to support claims about gender differences.

2-State explicitly whether this is a purely descriptive case-study or part of a mixed-methods design. If it is hypothesis‐driven, articulate the specific hypotheses being tested (e.g., “women in pSTEM will show shorter promotion intervals than women in STEM”).

3-Provide a flowchart or table listing each data source, date accessed, query parameters (e.g., “all personnel records with active status as of July 1, 2023”), and any filters applied (e.g., exclusion of emeritus staff until 2018).

4-By expanding the Methods to include precise search protocols, clear variable definitions, inferential and multivariable analyses, and transparent code/ethics statements, the authors will substantially improve the study’s credibility, reproducibility, and impact.

**Do you want your identity to be public for this peer review?** For information about this choice, including consent withdrawal, please see our Privacy Policy

Reviewer #1: No

Reviewer #2: **Yes: ** Muhammad Shahzad Aslam

---

## [Author Response · Author response to Decision Letter 1]

22 Aug 2025

A Word document containing our detailed responses to the reviewers’ comments has been uploaded to the system.

---

## [Decision Letter · Decision Letter 1]

23 Sep 2025

Feminization of the precarious at the UNAM: examining obstacles to gender equality

PONE-D-25-15103R1

Dear Dr. Espinosa-Asuar,

We’re pleased to inform you that your manuscript has been judged scientifically suitable for publication and will be formally accepted for publication once it meets all outstanding technical requirements.

Kind regards,

Alejandro Botero Carvajal, Ph.D

Academic Editor

PLOS ONE

Additional Editor Comments (optional):

Reviewer #1:

Reviewers' comments:

Reviewer's Responses to Questions

**Comments to the Author**

Reviewer #1: All comments have been addressed

2. Is the manuscript technically sound, and do the data support the conclusions?

Reviewer #1: Yes

3. Has the statistical analysis been performed appropriately and rigorously?

Reviewer #1: Yes

4. Have the authors made all data underlying the findings in their manuscript fully available?

Reviewer #1: Yes

5. Is the manuscript presented in an intelligible fashion and written in standard English?

Reviewer #1: Yes

Reviewer #1: The author have addressed my concerns and questions on the previous draft. I enjoyed reading this version of the paper.

**Do you want your identity to be public for this peer review?** For information about this choice, including consent withdrawal, please see our Privacy Policy

Reviewer #1: No

---

## [Editor Report · Acceptance letter]

PONE-D-25-15103R1

PLOS ONE

Dear Dr. Espinosa-Asuar,

I'm pleased to inform you that your manuscript has been deemed suitable for publication in PLOS ONE. Congratulations! Your manuscript is now being handed over to our production team.

Kind regards,

on behalf of

Dr. Alejandro Botero Carvajal

Academic Editor

PLOS ONE